# Optimization of Ultrasonic-Assisted Bioactive Compound Extraction from Green Soybean (*Glycine max* L.) and the Effect of Drying Methods and Storage Conditions on Procyanidin Extract

**DOI:** 10.3390/foods11121775

**Published:** 2022-06-16

**Authors:** Rattanaporn Khonchaisri, Nutsuda Sumonsiri, Trakul Prommajak, Pornchai Rachtanapun, Noppol Leksawasdi, Charin Techapun, Siraphat Taesuwan, Anek Halee, Rojarej Nunta, Julaluk Khemacheewakul

**Affiliations:** 1Division of Food Science and Technology, School of Agro-Industry, Faculty of Agro-Industry, Chiang Mai University, Chiang Mai 50100, Thailand; rattanaporn.khon@gmail.com (R.K.); siraphat.t@cmu.ac.th (S.T.); 2Department of Agro-Industrial, Food and Environmental Technology, Faculty of Applied Science, King Mongkut’s University of Technology North Bangkok, Bangkok 10800, Thailand; nutsuda.s@sci.kmutnb.ac.th; 3Division of Food Safety, School of Agriculture and Natural Resources, University of Phayao, Phayao 56000, Thailand; tpromjak@gmail.com; 4Division of Packaging Technology, School of Agro-Industry, Faculty of Agro-Industry, Chiang Mai University, Chiang Mai 50100, Thailand; pornchai.r@cmu.ac.th; 5Center of Excellence in Materials Science and Technology, Faculty of Science, Chiang Mai University, Chiang Mai 50200, Thailand; noppol.l@cmu.ac.th; 6The Cluster of Agro Bio-Circular-Green Industry (Agro BCG), Faculty of Agro-Industry, Chiang Mai University, Chiang Mai 50100, Thailand; 7Bioprocess Research Cluster, School of Agro-Industry, Faculty of Agro-Industry, Chiang Mai University, Chiang Mai 50100, Thailand; charin.t@cmu.ac.th; 8Cluster of Innovative Food and Agro-Industry, Faculty of Agro-Industry, Chiang Mai University, Chiang Mai 50100, Thailand; 9Division of Food Science and Technology, Faculty of Science and Technology, Kamphaeng Phet Rajabhat University, Kamphaeng Phet 62000, Thailand; nek_ha@hotmail.co.th; 10Division of Food Science and Technology, Faculty of Science and Technology, Lampang Rajabhat University, Lampang 52100, Thailand; quan_rn@hotmail.com

**Keywords:** procyanidin, green soybean, ultrasonic extraction, bioactive compounds, drying process, storage condition

## Abstract

Green soybean (*Glycine max* L.) seeds (GSS) are rich in various antioxidants and phytonutrients that are linked to various health benefits. Ultrasound-assisted extraction (UAE) technology was used for extracting the effective components from GSS. A response surface method (RSM) was used to examine the influence of liquid-to-solid ratio and extraction temperature on the bioactive compounds and antioxidant characteristics. The optimal conditions were a liquid-to-solid ratio of 25:1 and a UAE temperature of 40 °C. The observed values coincided well with the predicted values under optimal conditions. Additionally, the effects of drying methods on the procyanidins and antioxidant activities of GSS extract were evaluated. The spray-dried GSS extract contained the highest levels of procyanidins (21.4 ± 0.37 mg PC/g), DPPH (199 ± 0.85 µM Trolox eq/g), and FRAP (243 ± 0.26 µM Trolox eq/g). Spray drying could be the most time- and energy-efficient technique for drying the GSS extract. The present study also assessed the effects of storage temperature and time on procyanidins and antioxidant activities in GSS extract powder. Procyanidins were found to degrade more rapidly at 45 °C than at 25 °C and 35 °C. Storage under 25 °C was appropriate for maintaining the procyanidin contents, DPPH, and FRAP activities in the GSS extract powder. This study contributed to the body of knowledge by explaining the preparation of procyanidin extract powder from GSS, which might be employed as a low-cost supply of nutraceutical compounds for the functional food industry and pharmaceutical sector.

## 1. Introduction

Green soybean (*Glycine max* L.) seeds (GSS) are an excellent source of both nutritional and non-nutrient components useful to the human body. Among the many functional features of legumes, antioxidant activity stands out [1]. Generally, the antioxidant capacity of bean seeds is due to their high phenolic content. These are natural antioxidants that defend against reactive oxygen species that are accountable for the mechanisms that underpin the development of many serious diseases. Consumption of beans has been associated with a lower risk of atherosclerosis, colon cancer, coronary heart disease, diabetes, gastrointestinal diseases, and obesity [2]. Antioxidants have a well-established role in preventive medicine. A surge in interest in the discovery of novel, safe, non-toxic, and valuably effective organic antioxidants has occurred as a result of their numerous benefits, such as anti-aging and anti-inflammatory properties [3].

Plant procyanidins are secondary metabolites present in a variety of plant-based foods and drinks. They are an ubiquitous category of polyphenol polymers of flavan-3-ol [4]. Procyanidins have a variety of chemical and biological functions, including ultraviolet absorption, antimicrobial, anti-inflammatory, anti-allergic, and antioxidant properties. A study has determined that higher procyanidin consumption is related to a lower cancer risk. Based on their health advantages, procyanidins are potentially beneficial as dietary antioxidants and nutraceuticals [3]. Typically, procyanidins are derived from several natural sources, including apples, cinnamon, grape peels, grape seeds, legumes, and maritime pine bark [5].

Several extraction techniques have been used in the past to extract procyanidins from plants, including heating reflux extraction and homogenate extraction using water, methanol, ethanol, acetone, acetic ether, and other mixtures as solvents. However, it has been observed that the primary drawbacks of heating reflux extraction and homogenate extraction are the long extraction durations and low yields owing to repeated distillation, which prolongs the heating time and promotes oxidation of the extract [6,7]. Moreover, because of their toxicity, volatility, and flammability, organic solvents are troublesome for procyanidin extraction [5]. Following the introduction of “Green Chemistry,” ecologically friendly solutions have become appealing for overcoming the aforementioned challenges. The applications of ultrasound-assisted extraction (UAE) have received much more attention. UAE could extract bioactive components in a fraction of the time, at a lower temperature, and with less energy and solvent usage [8]. Saifullah et al. [9] compared the extraction efficiency of the UAE technique to that of the conventional shaking water bath method. The UAE technique was found to be more efficient in the extraction of TPC, TFC, proanthocyanidins, and antioxidant properties from lemon-scented tea tree (*Leptospermum petersonii*) leaves. The TPC value using UAE was significantly higher (15.27%) than the value obtained by shaking water bath extraction. UAE is a well-established technology for processing plant material and extracting analytes from diverse plant sections that may be readily scaled up for commercial production and could provide high repeatability in a shorter time frame with simplified manipulation [5].

Drying is a technique that removes moisture from fresh materials and reduces their water activity, thus further inhibiting microbial growth and minimizing deteriorative biochemical reactions. It also minimizes the weight and volume of the sample, thereby lowering storage and transportation expenses [10]. Various drying approaches, such as freeze drying, hot-air drying, vacuum drying, and spray drying, as well as drying conditions including time, temperature, and air velocity, have been linked to specific energy consumption and large impacts on the phytochemical compounds and antioxidant properties of the samples [11]. Furthermore, it is important to determine the optimal method for storing a particular type of bioactive compound that has been extracted.

The aims of this study were to optimize the UAE extraction conditions (water to sample ratio and temperature) of bioactive compounds from GSS using RSM with the Design-Expert 6.0.11 software. In addition, the effects of various drying techniques (freeze drying, hot-air drying, spray drying, and vacuum drying) and storage conditions were compared based on procyanidins of green soybean extract.

## 2. Materials and Methods

### 2.1. Materials

Fresh GSS were obtained from Lanna Agro Industry Co., LTD. (LACO, Chiang Mai, Thailand). After soaking the seeds in tap water for 1 min, they were oven-dried at 60 °C for 48 h in a hot air oven (Memmert UF 110, Schwabach, Germany) until the moisture content fell below 10% [12]. The dried seeds were ground into fine powder using the cyclotec sample mill (HR2602, Philips, Shanghai, China) with a sieve size of 40-mesh. GSS powder was packed in a vacuum aluminum foil bag and stored at a refrigerated temperature (3–5 °C) until further analysis. All compounds were of analytical quality.

### 2.2. Ultrasound-Assisted Extraction of Procyanidins

Extractions using water as solvent were carried out in an ultrasonic probe (VX500, Newtown, CT, USA) at 50% amplitude for 13 min with a maximum power of 500 W at 20 kHz frequency [1]. The effects of different liquid-to-solid ratios (15:1, 20:1, and 25:1 mL/g) and extraction temperature levels (40 °C, 50 °C, and 60 °C) on the total phenolic contents (TPC), total flavonoid contents (TFC), procyanidin content, and antioxidant activities were determined. After extraction, the mixtures were centrifuged at 5000 RPM for 15 min at 4 °C (Nüve NF400R, Ankara City, Turkey), and the supernatant was filtered through filter paper (Whatman No. 1, Wallingford, UK). The filtered extracts were obtained in a centrifuge tube and stored at −18 °C until further investigation.

### 2.3. Drying Conditions

The impacts of various drying techniques were evaluated using four distinct approaches: hot air, spray, vacuum, and freeze drying. Maltodextrin (4.0–7.0 DE) was applied as a coating agent after extraction [13]. A mixture of maltodextrin and extract was obtained by mixing 5 g of maltodextrin with 100 mL of GSS aqueous extract. It was then heated at 60 °C for 1 h before being exposed to ultraturax at 11,000 rpm for 5 min.

Freeze drying: GSS extract was spread to a layer thickness of roughly 1 cm on an aluminum tray and maintained at −18°C in a deep freezer for 24 h. Lyophilization of samples was carried out in a freeze dryer (LABCONCO, Kansas, MO, USA) to a constant weight for 42 h. Throughout the drying operation, the vacuum chamber pressure and condenser temperature were kept at 0.133 mbar and −40 °C, respectively.

Hot air drying: 250 g of GSS extract were spread on two separate 38 × 89 × 9 cm^3^ trays. The samples were dried to a constant weight and moisture content lower than 10% under drying temperatures of 50 °C using a hot air convection oven (FED 53, Binder, Tuttlingen, Germany) for 25 h.

Spray drying: 250 g of GSS extract was spray-dried for 6 h in a BUCHI Mini Spray Dryer (B-290, Flawil, Switzerland) under the following conditions: drying air inlet temperature of 130 °C; outlet temperature of 48 °C; feed volumetric flow rate of 2 mL/min; aspirator rate of 50%; pump rate of 5%; and cleaning needle for a nozzle of 0.3 mm diameter. The dispersing nozzle with a diameter of 0.7 mm was used.

Vacuum drying: 250 g of GSS extract was evaporated with water by a rotary evaporator (R-200, BUCHI) at 45 °C, 72 mbar for 3 h or until the sample was slurry. Samples were dried in a vacuum dryer (VD 53, Binder, Tuttlingen, Germany) at a drying temperature of 55 °C with a constant vacuum pressure (0 mbar) until a constant weight was reached after 20 h.

The drying time was observed after drying. Dehydrated samples were put into vacuum aluminum foil bags and refrigerated until further use.

### 2.4. Energy Consumption of Drying Methods

The energy consumption of a freeze dryer, hot air oven, spray dryer, and vacuum dryer were calculated using Equation (1) as described by Nguyen et al. [11].
(1)$$\mathrm{EC}=\frac{D_{temp}}{M_{temp}}\;\times MO$$
where EC was the energy consumption (kWh), *D_temp_* was the drying temperature used (°C), *M_temp_* was the maximum temperature (°C) of the drying equipment, *MO* was the maximum energy output (kW) for the drying equipment, and T was the drying duration (h). The energy consumption by the freeze dryer at various time periods was estimated using Equation (2) as described by Nguyen et al. [14].
(2)EC=P×t
where P denoted the supplied electrical power (kW) and t denoted the time required to dry the sample (h).

### 2.5. Storage Stability Test of GSS Extract Powder

The GSS extract powder was transferred to an aluminum foil bag and vacuum sealed before being stored at regulated temperatures of 25 °C, 35 °C, and 45 °C. Their individual procyanidin and antioxidant activities were determined for up to 28 days at specified intervals (day 0, 7, 14, 21, and 28).

### 2.6. Determination of Phytochemicals and Antioxidant Properties

Approximately 1 g of samples was mixed with 10 mL of 80% acetone and agitated for 15 min. The samples were then homogenized for 15 min at 30 °C followed by 15 min of shaking. After 15 min of centrifuging the samples at 10,000 rpm, the supernatants were collected and analyzed.

#### 2.6.1. Phytochemical Analysis

##### Total Phenolic Contents (TPC)

An aliquot of the supernatant (approximately 1 mL) was diluted with water to approximately 7 mL, and 0.5 mL of Folin-Ciocalteau phenol reagent was added. 1.0 mL of saturated sodium carbonate solution was added exactly 5 min later. The mixture was then diluted with water to a volume of 10 mL. The samples were incubated at room temperature for 2 h before the absorbance at 765 nm was measured with a UV spectrophotometer (Cary 60 Bio, UV-Vis, Kuala Lumpur, Malaysia). The standard curve was constructed using gallic acid and the results were represented as mg of gallic acid equivalent per g of dry weight sample (mg GAE/g dw). The calibration curve equation was Y = 10.951X + 0.0143 while the determination coefficient was R^2^ = 0.999, where Y denoted light absorbance and X denoted compound concentration.

##### Total Flavonoid Contents (TFC)

The TFC of GSS extract was determined using a previously reported technique [15]. The absorbance was measured at 510 nm using a UV spectrophotometer (Cary 60 Bio, UV-Vis, Kuala Lumpur, Malaysia). The standard curve was constructed using catechin and the results were expressed as mg of catechin equivalents per g of dry weight sample (mg CE/g dw). The calibration curve equation was Y = 3.7931X + 0.0295 while the determination coefficient was R^2^ = 0.99384, where Y denoted light absorbance and X denoted compound concentration.

##### Determination of Procyanidins

The procyanidin concentration was measured using a modified vanillin-sulfuric acid technique [16]. Standard solutions containing 0, 0.025, 0.05, 0.075, 0.1, 0.125, and 0.15 mg/mL of procyanidin (PC) were prepared. A 1 mL portion was transferred to separate test tubes where it was mixed with 3 mL of 30% sulfuric acid/methanol solution and 6 mL of 30 g/mL vanillin/methanol solution. The mixture was then held at 30 °C for 30 min. A UV-visible spectrophotometer (UV9600, Bobang Co., Zhengzhou, China) was used to measure the absorbance at 500 nm, and the standard PC absorption curve was drawn. The sample was then treated in the same manner. The procyanidin concentration was reported as mg per g of dry weight of GSS extract (mg PC/g).

#### 2.6.2. Antioxidant Analysis

##### DPPH Radical Scavenging Capacity Assay

The ability of GSS extract to scavenge DPPH radicals was assessed using the method described by Thaipong et al. [17]. The absorbance at 515 nm was measured using a UV spectrophotometer (Cary 60 Bio, UV-Vis, Kuala Lumpur, Malaysia). The calibration curve was developed using Trolox, and the measurements were represented as µmol of Trolox equivalents per g of dry weight sample (µM Trolox eq/g). The equation for the calibration curve was Y = 0.0025X-0.031 and the determination coefficient was R^2^ = 0.9597, where Y denoted light absorbance and X denoted compound concentration.

##### Ferric-Reducing Antioxidant Power (FRAP) Assay

The FRAP of GSS extract was assessed following the procedure described by Thaipong et al. [17]. The absorbance at 593 nm was measured using a UV spectrophotometer (Cary 60 Bio, UV-Vis, Kuala Lumpur, Malaysia). Trolox was used to develop a calibration curve and the results were represented as µmol of trolox equivalents per g of dry weight sample (µM Trolox eq/g). The equation for the calibration curve equation was Y = 0.0062X + 0.0099 and the determination coefficient was R^2^ = 0.997, where Y denoted light absorbance and X denoted compound concentration.

### 2.7. Experimental Design and Statistical Analysis

Relationships between extraction factors (liquid-to-solid ratio and extraction temperature) and response variables were fitted with a quadratic model using Design-Expert 6.0.11 (Stat-Ease, Minneapolis, MN, USA). The models were optimized for the maximum value of all response variables. Comparisons among different drying conditions and storage stability tests on procyanidin content, as well as antioxidant activities, were analyzed using one-way analysis of variance. Duncan’s new multiple range post-hoc analysis was used to detect statistically significant differences (*p* < 0.05) between samples. SPSS for Windows version 16 was used for all the aforementioned analyses. The data was presented in the form of mean values and standard deviations.

## 3. Results

### 3.1. Optimization Process Conditions for UAE Extraction of Bioactive Compounds from GSS

To determine the impact of the factors and their variation, the effects of extraction temperature and liquid-to-solid ratio on the concentration of bioactive compounds from GSS were investigated. As shown in Table 1, the highest concentration (*p* < 0.05) of TPC (76.8 ± 0.46 mg GAE/g), TFC (30.9 ± 0.58 mg CAE/g), procyanidins (21.1 ± 0.05 mg PC/g) and antioxidant activities of DPPH (192 ± 0.09 µM Trolox eq/g) and FRAP (318± 0.71 µM Trolox eq/g) were attained under the extraction condition of 40 °C and liquid-to-solid ratio of 25:1. All bioactive compounds had rather high content at the lowest extraction temperature of 40 °C, which gradually decreased as the UAE temperature increased from 50 °C to 60 °C. Because the pressure gradient between within and outside the bubble is minimized at high temperatures, the higher vapour pressure of the solvent occupies the cavitation bubble that occurs during the rarefaction cycle. As a result, even if the quantity of cavitation bubbles is great at high temperatures, they collapse with less intensity, resulting in less cell injury and a lower yield. This result was consistent with the findings obtained with proanthocyanidins from guava leaves [18]. The proanthocyanidin content decreases when the UAE temperature exceeds 50 °C [19]. According to Kumar et al. [8], increasing the temperature initially increases the yield of the UAE, but further increasing the temperature decreases the yield. Increased temperature reduces yield not only due to the weakened cavitation effect, but also due to the decomposition of bioactive compounds released into the extraction medium. UAE should be processed at a low operating temperature in order to maintain high extract quality for compounds [20]. The impact of heating temperature on polyphenol extraction from plant material is related to the various types and bound forms of polyphenol found in plants depending on the species [21]. As a consequence, different plant species produce optimal extraction and recovery rates of extracted chemicals. The ultrasonication temperature of 40 °C was the optimum temperature for the high yield of TPC, TFC, and procyanidin content, as well as the antioxidant capacity of the GSS extract.

It was also found that the recovery of TPC, TFC, and procyanidin content was parallel to the increase in liquid-to-solid ratio from 15:1 to 25:1 mL/g at the extraction temperature of 40 °C. Similar trends were observed in DPPH and FRAP values, where a liquid-to-solid ratio of 25:1 at the extraction temperature of 40 °C resulted in the highest antioxidant capacity (192 ± 0.09 and 318 ± 0.71 µM Trolox eq/g, respectively) of GSS extract. This was consistent with the mass transfer principle, which indicates that the concentration gradient between solid and liquid is the driving force during mass transfer and rises with increasing solvent-to-sample ratio [22]. Li et al. [3] found a favourable correlation between procyanidin concentration and liquid-to-solid ratio, which rose rapidly from 5:1 to 15:1 mL/g.

### 3.2. Response Surface Methodology (RSM) Analysis

Regression models for actual factors (where T was temperature and R was liquid-to-solid ratio) were obtained as followed:

Total phenolics = 634.08 − 22.59T − 1.33R + 0.22T^2^ + 0.14R^2^ − 0.06TR

Total flavonoids = 266.69 − 10.82T + 3.01R + 0.11T^2^ − 0.05TR

Procyanidin = 82.81 − 2.08T − 1.04R + 0.02T^2^ − 0.05R^2^ − 0.02TR

DPPH = 49.64 − 2.64T + 8.91T + 0.03T^2^ − 0.02TR

FRAP = 2497.37 − 92.62T + 15.31R + 0.88T^2^ − 0.27TR

Response surface plots of the response variables are shown in Figure 1. All models were significant (*p* < 0.05) with an R^2^ greater than 0.95, except for the procyanidin model (R^2^ = 0.70), as shown in Table 1. Although the ratio-containing terms in the procyanidin model had an insignificant *p*-value, they were included in the model to retain higher R^2^. The optimal extraction conditions for antioxidative compounds were 40 °C and a ratio of 1:25, which resulted in predicted values of 73.63 mg GAE/g total phenolics, 30.72 mg GAE/g total flavonoids, 19.03 mg GAE/g procyanidin, 191.31 µmoL Trolox eq/g DPPH, and 313.84 µmoL Trolox eq/g FRAP. The models were validated using optimal extraction conditions. The actual response values were not significantly different from those predicted values, indicating the suitability of the models (Table 2 and Table 3). The results implied that the experimental value was consistent with the predicted values. Not only were the extraction conditions established by RSM precise and trustworthy, but they also had practical value.

### 3.3. Effect of Drying Methods on Procyanidin Content and Antioxidant Properties of GSS Extract

In this research, four different procedures were applied to dry GSS extract in order to determine the effect of different drying methods on procyanidin content, antioxidant properties, and energy consumption. The spray drying process produced the highest levels of procyanidin content (21.4 ± 0.37 mg PC/g) and FRAP (243 ± 0.26 µM Trolox eq/g) (Table 4). It was worthwhile to note that the quantities of procyanidins were not significantly different when the GSS extract was vacuum dried (*p* > 0.05). Our findings indicated that hot air and freeze-drying significantly affected the retention of procyanidins. These findings can be explained by the long exposure time of hot air drying for 25 h, resulting in the degradation of procyanidins. Vu et al. [23] obtained similar results during the drying of banana (*Musa cavendish*) peels. They found that drying at high temperatures for a shorter duration preserved more phytochemical concentrations and antioxidant properties than drying at lower temperatures for a longer duration.

In the case of freeze drying, GSS extract dried under this method contained less procyanidin than those dried via spray-drying and vacuum-drying processes. Similarly, Papoutsis et al. [24] reported that lemon pomace dried under vacuum had a higher total flavonoid content than did samples dried via freeze-drying and hot air-drying techniques. Buratto et al. [25] evaluated total phenolic and total flavonoid content preserved in the obtained powders of dehydrated Feijoa pulp (*Acca sellowiana*) after spray-drying and freeze-drying techniques. Spray drying was found to be more effective than freeze drying at preserving total phenolic and total flavonoid contents (*p* < 0.05), resulting in relatively higher contents of phenolic and total flavonoids (67.4 ± 4.70 mg GAE/g and 78.8 ± 6.56 mg QE/g, respectively) than the freeze-drying method (49.45 ± 0.01 mg GAE/g and 7.80 ± 0.82 mg QE/g, respectively). Saikia et al. [26] preserved phenolic compounds extracted from *Averrhoa carambola* pomace using spray-drying and freeze-drying methods. Their results indicated that the spray-drying method resulted in a greater concentration of phenolic compounds. Although the inlet temperature (130 °C) of the spray dryer equipment is high, the process installs almost instantaneously and, therefore, the losses of bioactive compounds due to drying are associated with the outlet temperature (48 °C), which is lower and has no effect on the degradation of phenolic compounds.

Regarding the freeze-drying method, the losses in this process are associated with the grinding of the material after the drying process. The lower concentrations of total phenolic compounds and total flavonoids in freeze-dried powder as compared to spray-dried powder may be attributable to the slower drying rate and longer drying time, which may contribute to unfavourable conditions for phenol composition, resulting in the destruction of the compounds [27]. The time consumption for the drying of samples by different mechanical techniques in the present study was as follows: freeze drying > hot air drying > vacuum drying > spray drying.

The antioxidant activity of the samples was assessed based on their procyanidin content, as shown in Table 4. The sample dried by the spray-drying method presented higher antioxidant activity of DPPH and FRAP. In general, it was observed that the freeze-dried powder showed lower antioxidant activity compared to the spray-dried powder, indicating that the method was efficient and satisfactory. Ramírez et al. [27] explained that the use of high temperatures in the spray-drying method presented a risk for the bioactive compounds. However, GSS is abundant in flavonoids, which are mainly present in glycoside forms such as procyanidins, quercetin, glycitein, and daidzein. Moreover, glycosylated flavonoids are more resistant to heat treatment than aglycon flavonoids. Hence, the use of high temperatures in the spray-drying method does not cause drastic degradation of procyanidins, contributing to the high antioxidant activity of the obtained powders [1,28].

Although freeze drying is considered a low nutrient degradation method due to the low temperature, some losses of bioactive compounds and volatile substances occur mainly during the sublimation stage of ice. In the sublimation phase, compounds with a higher vapour pressure than water molecules are excluded and evaporated from the frozen materials when the temperature of the sample matrix exceeds its glass transition temperature [25], leading to lower antioxidant activity when compared to the spray-drying method.

As it is directly proportional to the cost of drying, energy consumption is a crucial element in the selection of appropriate drying techniques. The energy consumption of the various drying methods was calculated to determine the differences in the amount of energy required to dry GSS extract using each mechanical drying technique (Table 4). The most energy-intensive drying method was freeze drying (184 kWh), followed by spray drying, vacuum drying, and hot air drying (7.88, 6.57, and 5.00 kWh, respectively). On the contrary, the hot-air- and vacuum-drying methods consumed less energy than spray drying but required longer time periods for drying (25 and 20 h, respectively). The continuous nature of spray drying has substantial economic advantages, with the cost of spray drying estimated to be approximately one-sixth that of freeze drying [29]. According to Chávez and Ledeboer [30], spray drying is the most widely used and studied alternative to freeze drying because it is cost-effective, easily available, simple to use, and scalable. Spray drying is four to seven times less expensive and more energy efficient than freeze drying. Moreover, spray drying has a fixed cost of 12% of that of freeze drying, and a manufacturing cost of 20% of that of freeze drying [31].

### 3.4. Effect of Storage Temperature and Time on the Stability of Retained Procyanidins of Dry GSS Extract

The storability and eventual quality of GSS extract powder were influenced by the optimality of the storage conditions. The present study assessed the effects of storage temperature (25 °C, 35 °C, and 45 °C) and time (0–28 days) on procyanidins and antioxidant activities of GSS extract powder. The investigated temperatures were chosen in order to estimate the stability of procyanidin enriched-extract during storage. The initial procyanidin content was 20.5–21.5 mg PC/g. The loss of procyanidins from GSS extract powder at the end of storage was 14.8 ± 0.40, 10.4 ± 0.24, and 7.56± 0.16 mg PC/g with the retention percentage of 68.6 ± 0.13, 49.6 ± 0.39, and 35.6 ± 0.47% at 25, 35 and 45 °C (Table 5), respectively. Although procyanidins of GSS extract powder decreased significantly (*p* < 0.05) after 14 days at the storage temperature of 25 °C (Figure 2A), 25 °C was more stable than other levels of temperature, maintaining 70% stability during 21 days of storage. In contrast, the maximum decrease in procyanidin content was observed at 45 °C, which decreased drastically after 21 days of storage. The results were consistent with Pavlović et al. [32], who reported that procyanidins in dark chocolate decreased at all storage temperatures (4–35 °C) with significant differences observed at *p* < 0.05. After 30 days of storage, procyanidin degradation was greater at 35 °C than at 4 °C and 22 °C.

Procyanidins may be lost during storage as a result of procyanidins becoming bonded to macromolecules such as proteins, and to polysaccharides such as maltodextrin during storage [33]. Based on the experiment conducted by Widyaningsih et al. [34], the decrease in flavonoid and antioxidant activity from black garlic (*Allium sativum* L.) aqueous extract powder was due to the relatively high amount of maltodextrin added. Antioxidant activities followed a similar trend to procyanidin content under the same storage conditions (Figure 2B,C). The DPPH and FRAP retention decreased to less than 83.9% at all levels of temperature after the end of storage (28 days).

In comparing the data to assess the impact of storage temperature in the present study, we found a significant difference (*p* < 0.05) across temperatures (25 °C, 35 °C, and 45 °C) during 7–28 days of storage (Figure 2A). Procyanidins of dried extract kept at the lowest temperature of 25 °C without exposure to light and air exhibited the greatest stability with over 68% retention throughout the entire storage period of 28 days. While the procyanidins of the extract stored at higher temperatures of 35 °C and 45 °C decreased rapidly, with a retention of 49.6 ± 0.39 and 35.6 ± 0.47%, respectively, at 28 days of storage. The degradation rate of procyanidins during storage was strongly impacted by temperature, which was fastest at 45 °C and slowest at 25 °C. According to the statement by Dallas et al. [35], procyanidins decreased 5 or 10 times faster at the higher temperatures of 32 and 42 °C. Mrmošanin et al. [36] observed a similar effect in cocoa powders, where the procyanidin concentration decreased as the temperature increased; specifically, the loss was greater than 50% after 30 days at 35 °C. Del-Toro-Sánchez et al. [37] reported a massive 36% reduction in the total phenolic compounds of the herb, *Anemopsis californica*, extracts stored at 50 °C, 25 °C, and 4 °C for 180 days. They confirmed that high temperatures degraded the structure of the polyphenols.

Storage temperature has been found to affect the stability of procyanidins as well as their corresponding antioxidant activities of DPPH and FRAP (Table 5). The decrease in antioxidant activity of GSS extract powder according to storage temperature was considered to be due to the decrease in procyanidin content. A significant decrease in DPPH and FRAP activities (*p* < 0.05) was observed at 25 °C (184 ± 0.83 and 239 ± 2.26 µM Trolox eq/g, respectively) and 35 °C (182 ± 2.29 and 244 ± 1.61 µM Trolox eq/g, respectively) until 14 days of storage. The authors further highlighted that more than 85% of the antioxidant activities were retained at 25 °C and 35 °C after 21 storage days. The lowest percentage retention of DPPH (67.4 ± 0.38%) and FRAP (74.1 ± 0.18%) were observed at 45 °C after 28 days, which correlated with the lowest percentage retention of procyanidins (35.6 ± 0.47%), indicating that the bioactive compounds degraded more rapidly at higher temperatures. Consistent with the findings of Zorić et al. [38], the antioxidant activity of Marasca sour cherry powder decreased significantly when stored at 37 °C compared to 4 °C. The antioxidant activity of GSS extract powder was stable at the recommended storage temperature and time, implying that it could be used as a functional food supplement.

## 4. Conclusions

In this research, bioactive compounds were extracted from GSS using an effective UAE approach, optimised using RSM. The optimal extraction conditions were a 25:1 liquid-to-solid ratio and a UAE temperature of 40 °C. Under these circumstances, the experimental yield of TPC (76.6 ± 0.54 mg GAE/g), TFC (28.9 ± 0.55 mg CAE/g), procyanidins (19.2 ± 0.33 mg PC/g), and antioxidant activities of DPPH (194 ± 3.53 µM Trolox eq/g) and FRAP (320 ± 4.88 µM Trolox eq/g) were correlated with the predicted values, indicating the capability of the model and the success of RSM in optimising the extraction conditions. The amount of procyanidin and anti-oxidant properties of GSS extract were affected by the manner of drying. Spray drying was the most effective in terms of the highest concentration level of procyanidins and antioxidant activity, as well as the lowest energy consumption. Hence, spray drying should be considered as a cost-effective drying method for large-scale production with the added benefit of maximum GSS extract recovery. The dried extract stored at low temperatures of 25°C showed the highest retention of procyanidins, with equally good corresponding DPPH and FRAP antioxidant activity after a storage time of 28 days. Finally, this study suggested that dried GSS extract powder could be developed as a potential natural antioxidant for the food industry and pharmaceutical sector. Further research is necessary to determine the potential prevention and treatment of oxidative stress-induced diseases by purified procyanidins from GSS.

## Figures and Tables

**Figure 1 foods-11-01775-f001:**
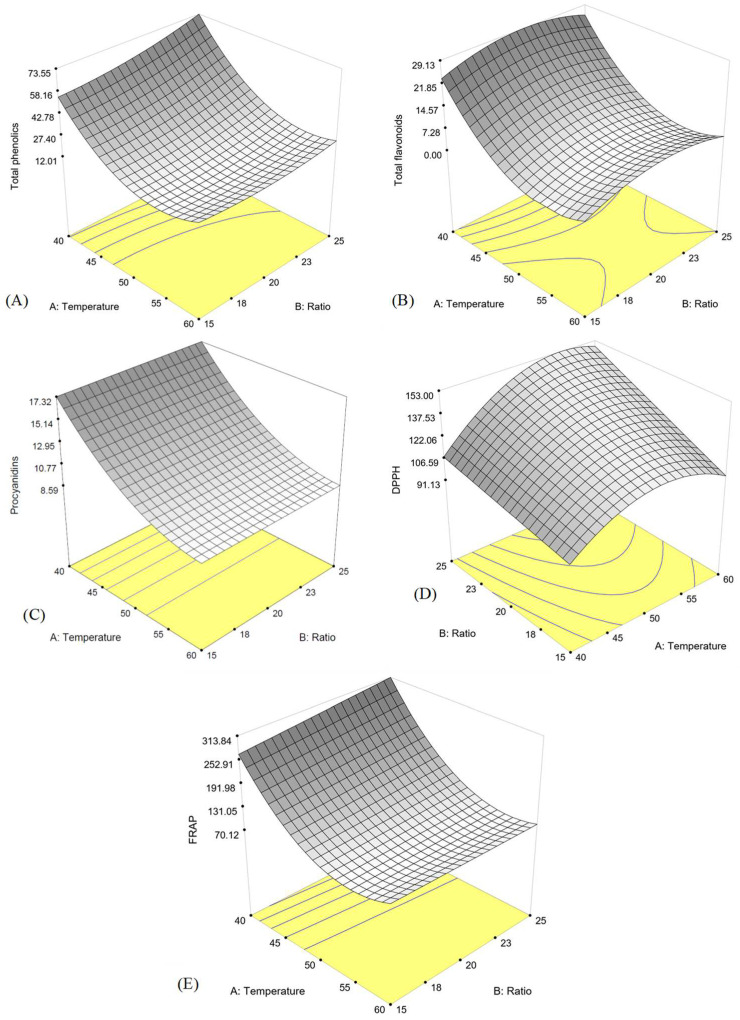
Response surface plot of (**A**) TPC, (**B**) TFC, (**C**) procyanidins, (**D**) DPPH and (**E**) FRAP as a function of extraction temperature and solvent-to-solid ratio.

**Figure 2 foods-11-01775-f002:**
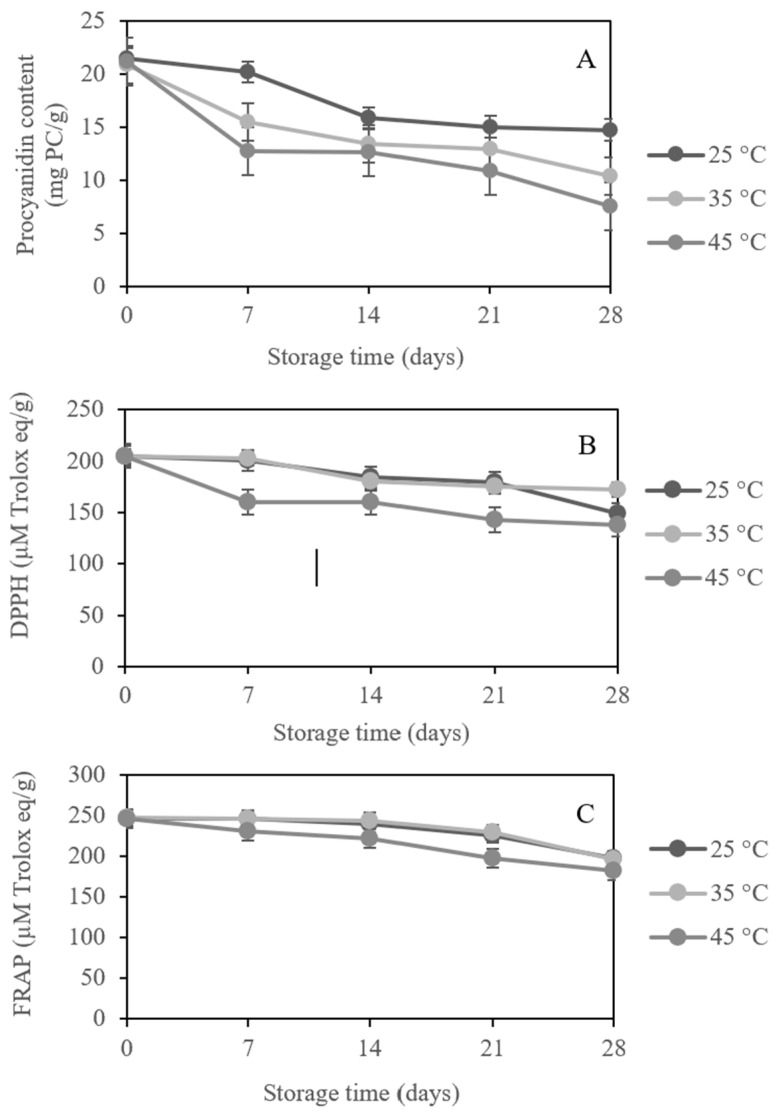
Changes in the procyanidins (**A**), DPPH radical scavenging activity (**B**), and FRAP radical scavenging activity (**C**) of GSS extract powder over 28 days at various storage temperatures.

**Table 1 foods-11-01775-t001:** Effects of liquid-to-solid ratio and extraction temperature on bioactive compounds and antioxidant capacity of GSS extract.

Temperature (°C)	Liquid-to-Solid Ratio	Bioactive Compounds			Antioxidant Activity	
		TPC (mg GAE/g)	TFC (mg CAE/g)	Procyanidins (mg PC/g)	DPPH (µM Trolox eq/g)	FRAP (µM Trolox eq/g)
	15:1	55.2 ± 0.45 ^b^	18.2 ± 0.73 ^c^	14.6 ± 0.09 ^d^	112 ± 0.05 ^g^	255 ± 0.41 ^c^
40	20:1	56.2 ± 0.63 ^b^	28.6 ± 0.67 ^b^	16.2 ± 0.07 ^b^	150 ± 0.05 ^d^	299 ± 0.28 ^b^
	25:1	76.8 ± 0.46 ^a^	30.9 ± 0.58 ^a^	21.1 ± 0.05 ^a^	192 ± 0.09 ^a^	318 ± 0.71 ^a^
	15:1	17.1 ± 0.45 ^e^	6.09 ± 0.10 ^d^	15.8 ± 0.09 ^c^	106 ± 0.27 ^i^	107 ± 0.38 ^d^
50	20:1	17.4 ± 0.28 ^e^	3.86 ± 0.11 ^e^	8.54 ± 0.07 ^e,f^	145 ± 0.10 ^f^	105 ± 0.46 ^e^
	25:1	26.6 ± 0.38 ^c^	4.06 ± 0.06 ^e^	8.39 ± 0.03 ^e,f^	186 ± 0.07 ^b^	92.7 ± 0.33 ^f^
	15:1	15.2 ± 0.96 ^f^	3.55± 0.03 ^e^	6.88 ± 0.04 ^f^	108 ± 0.09 ^h^	81.8 ± 0.89 ^g^
60	20:1	23.9 ± 0.47 ^d^	5.39 ± 0.07 ^d^	9.43 ± 0.03 ^e^	149 ± 0.10 ^e^	92.1 ± 0.6 ^f^
	25:1	25.1 ± 0.97 ^c,d^	6.07 ± 0.06 ^d^	9.45 ± 0.03 ^e^	184 ± 0.03 ^c^	91.7 ± 0.48 ^f^

Data are expressed as means ± standard deviation (*n* = 4). Different letters (a–i) in the same column represent statistically significant difference (*p* < 0.05).

**Table 2 foods-11-01775-t002:** Parameters of regression model for actual factors.

Parameters	Total Phenolics		Total Flavonoids		Procyanidin		DPPH		FRAP	
	Coefficient	*p*-Value	Coefficient	*p*-Value	Coefficient	*p*-Value	Coefficient	*p*-Value	Coefficient	*p*-Value
Model										
Constant	634.08		266.69		82.81		49.64		2497.37	
Temperature	−22.59	<0.001	−10.82	<0.001	−2.08	<0.001	−2.64	<0.001	−92.62	<0.001
Ratio	−1.33	<0.001	3.01	0.001	−1.04	0.617	8.91	<0.001	15.31	<0.001
Temperature^2^	0.22	<0.001	0.11	<0.001	0.02	0.046	0.03	<0.001	0.88	<0.001
Ratio^2^	0.14	0.006			−0.05	0.186				
Temp × Ratio	−0.06	0.002	−0.05	<0.001	−0.02	0.156	−0.02	<0.001	−0.27	<0.001
*p*-value	<0.0001		<0.0001		<0.0001		<0.0001		<0.0001	
R^2^	0.9785		0.9523		0.7005		0.9991		0.9871	

**Table 3 foods-11-01775-t003:** Predicted and actual response values obtained at optimal extraction conditions (40 °C and a liquid-to-solid ratio of 25:1).

Response Variables	Predicted Values	Actual Values	*p*-Value
Total phenolics (mg GAE/g sample)	73.6	76.6 ± 0.54	0.16
Total flavonoids (mg catechin/g sample)	30.7	28.9 ± 0.55	0.37
Procyanidin (mg/g sample)	19.0	19.2 ± 0.33	0.89
DPPH (µg Trolox eq/g sample)	191	194 ± 3.53	0.79
FRAP (µmol Trolox eq/g sample)	314	320 ± 4.88	0.72

**Table 4 foods-11-01775-t004:** Effects of different drying methods on extractable procyanidin contents and antioxidant properties.

Drying Method	Procyanidins (mg PC/g)	DPPH (µM Trolox eq/g)	FRAP (µM Trolox eq/g)	Energy Consumption (kWh)
Freeze drying	17.3 ± 0.14 ^b^	198 ± 0.65 ^a^	238 ± 0.3 ^b^	184
Hot air drying	12.1 ± 0.51 ^c^	127 ± 0.48 ^b^	122 ± 0.31 ^d^	5.00
Spray drying	21.4 ± 0.37 ^a^	199 ± 0.85 ^a^	243 ± 0.26 ^a^	7.88
Vacuum drying	20.8 ± 0.23 ^a^	200 ± 0.85 ^a^	207 ± 0.19 ^c^	6.57

Data are expressed as means ± standard deviation (*n* = 4). Different letters (a–d) in the same column represent statistically significant difference (*p* < 0.05).

**Table 5 foods-11-01775-t005:** Effect of storage temperature and time on the retention percentage of procyanidins and antioxidant activities of DPPH and FRAP in dry GSS extract powder.

Storage Temperature (°C)	Time (Days)	% Retention		
		Procyanidins	DPPH	FRAP
	0	100	100	100
	7	94.1 ± 0.38	97.9 ± 0.19	99.2 ± 1.42
25	14	73.8 ± 1.22	90.0 ± 0.51	97.4 ± 0.40
	21	70.0 ± 0.15	87.5 ± 1.40	91.9 ± 0.11
	28	68.6 ± 0.13	72.7 ± 0.89	80.2 ± 0.35
	0	100	100	100
	7	74.2 ± 0.16	98.9 ± 0.38	99.8 ± 0.17
35	14	64.2 ± 1.13	88.2 ± 1.66	98.9 ± 1.03
	21	61.9 ± 1.65	85.4 ± 1.61	92.9 ± 0.99
	28	49.6 ± 0.39	83.9 ± 0.70	79.5 ± 0.89
	0	100	100	100
	7	60.3 ± 0.85	78.1 ± 0.83	93.8 ± 0.29
45	14	59.8 ± 0.93	77.8 ± 0.26	90.4 ± 1.17
	21	51.5 ± 0.39	69.6 ± 0.32	80.1 ± 0.13
	28	35.6 ± 0.47	67.4 ± 0.38	74.1 ± 0.18

Data are expressed as means ± standard deviation (*n* = 4).

## Data Availability

The data presented in this study are available on request from the corresponding author.

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
