# Peer review of "Optimization of Ultrasonic-Assisted Bioactive Compound Extraction from Green Soybean (Glycine max L.) and the Effect of Drying Methods and Storage Conditions on Procyanidin Extract"

_foods, 2022, doi:10.3390/foods11121775_

Round 1
Reviewer 1 Report
Review of foods-1769655
This manuscript deals with the ultrasound-assisted extraction of green soybean, continued with spray drying, for procyanidin as the molecule of interest. The experiments and characterizations have been well conducted. However, there are several issues that must be clarified as follows:
1. Please make a list of abbreviations.
2. Please use the proper degree sign °C, not with superscripted zero, superscripted lowercase o, or superscripted uppercase O. --> Location: Line 32, 39, 39, 39, 98, 109, 111, 114, 121, 123, 126, 129, 133, 133, 137, 149, 149, 165, 193, 233, 235, 235, 242, 251, 256, 257, 318, 320, 370, 375, 377, 377, 379, 381, 383, 383, 394, 395, 397, 400, 400, 401, 404, 406, 411, 412, 414, 416, 420, 420, 447.
3. Line 27: Scientific names must be written in italic.
4. Line 38: Please change “pow-der” to “powder”.
5. Line 59-89: Please split this long paragraph to be 2-3 shorter paragraphs.
6. Line 61: …have a variety of…
7. Line 117-142: Please add references about the drying conditions. Besides ultrasonic-assisted extraction, there is also ultrasonic-assisted drying. Related to it for the drying of food and bioproducts, there is a recent review about it to add more references from the recent five years (2018-2022): Drying Technology 39 (2021) 1554-1576 https://doi.org/10.1080/07373937.2021.1914078
8. Line 134: What does it mean with “nozzle cleaner no. 4”? Please write the detail of the diameter or the length, or it would be even better if a figure is provided.
9. Line 147: Please add equation number next to the equation. Please delete the cross sign, because the equation is not vector operation
10. Line 153: Please add equation number next to the equation. Please delete the cross sign, because the equation is not vector operation
11. Line 228: …the impact of the factors…
12. Table 2: Please add the P-values for each parameter (temperature, L/S ratio, temperature × temperature interaction, ratio × ratio interaction, temperature × ratio interaction). The P-values of them are important to comprehend which parameter is significant, or whether the interaction is significant or not. As the significant parameters are known, the insignificant parameter(s) can be removed from the regression equation, and therefore simplifying the regression equation.
13. Table 2: Where are the regression equations? --> for (1) total phenolic, (2) flavonoids, (3) procyanidin, (4) DPPH, and (5) FRAP??
14. Table 2 and Table 3: For the regression equation, the parameters are in coded units (-1, 0, +1) or in real values (such as temperature 40, 50, 60 °C, and not -1, 0, +1, respectively)?
15. Line 293-346: Please split this very long paragraph (takes up the whole page) to 4-6 shorter paragraphs.
16. Line 368-391: Please split this long paragraph to two shorter ones.
17. Line 378: Change “Contrariwise” to “In contrast”, or “On the contrary”, or “On the other hand”.
18. Line 392-423: Please split this long paragraph to two shorter ones.
19. Line 454: Scientific names must be written in italic.
20. Line 513: Scientific names must be written in italic.
21. Line 545: Scientific names must be written in italic. Write the genus Anemopsis with uppercase A
Author Response
Please see the attachment
Response to Reviewer 1 Comments
The authors thank Reviewer 1 for his/her kind and constructive comments to improve the quality and clarity of our manuscript.
Point 1: Please make a list of abbreviations.
Response 1: The “Abbreviations list” was added after “Conflicts of Interest” as follows:
GSS green soybean seeds
UAE ultrasound-assisted extraction
RSM response surface method
TPC total phenolic contents
TFC total flavonoid contents
PC procyanidin
FRAP ferric-reducing antioxidant power
Point 2: Please use the proper degree sign °C, not with superscripted zero, superscripted lowercase o, or superscripted uppercase O. --> Location: Line 32, 39, 39, 39, 98, 109, 111, 114, 121, 123, 126, 129, 133, 133, 137, 149, 149, 165, 193, 233, 235, 235, 242, 251, 256, 257, 318, 320, 370, 375, 377, 377, 379, 381, 383, 383, 394, 395, 397, 400, 400, 401, 404, 406, 411, 412, 414, 416, 420, 420, 447.
Response 2: We rechecked and changed the proper degree sign °C throughout the manuscript.
Point 3: Line 27: Scientific names must be written in italic.
Response 3: We made a correction at Line 57 from “(Glycine max L.)” to “(Glycine max L.)”.
Point 4: Line 38: Please change “pow-der” to “powder”.
Response 4: We made a correction at Line 44 from “pow-der” to “powder”
Point 5: Line 59-89: Please split this long paragraph to be 2-3 shorter paragraphs.
Response 5: This long paragraph is splited into 3 paragraphs as follows:
Plant procyanidins,….
Several extraction techniques….
Drying is a technique….
Point 6: Line 61: …have a variety of…
Response 6: The phrase “...have variety of …” is changed to “have a variety of”
Point 7: Line 117-142: Please add references about the drying conditions. Besides ultrasonic-assisted extraction, there is also ultrasonic-assisted drying. Related to it for the drying of food and bioproducts, there is a recent review about it to add more references from the recent five years (2018-2022): Drying Technology 39 (2021) 1554-1576 https://doi.org/10.1080/07373937.2021.1914078
Response 7: The additional reference about the drying conditions is given in the sentence as following “The impacts of various drying techniques were evaluated using four distinct approaches: hot air, spray, vacuum, and freeze drying. Maltodextrin (4.0–7.0 DE) was applied as a coating agent after extraction [13].”
Point 8: Line 134: What does it mean with “nozzle cleaner no. 4”? Please write the detail of the diameter or the length, or it would be even better if a figure is provided.
Response 8: The nozzle cleaner is the nozzle-cleaning device that permits automatic or manual cleaning, which allows the manual insertion of a needle through the nozzle to prevent blockages. Hence, further explanation is given in the sentence as follows:
“…cleaning needle for a nozzles of 0.3 mm diameter”
Point 9: Line 147: Please add equation number next to the equation. Please delete the cross sign, because the equation is not vector operation
Response 9: The equation number (1) is added next to the equation. In addition, the cross sign contained in the equations is changed to the multiplication sign.
Point 10: Line 153: Please add equation number next to the equation. Please delete the cross sign, because the equation is not vector operation
Response 10: The equation number (2) is added next to the equation. In addition, the cross sign contained in the equations is changed to the multiplication sign.
Point 11: Line 228: …the impact of the factors…
Response 11: The phrase “To determine the impact factors.…” is changed to “To determine the impact of the factors….”
Point 12: Table 2: Please add the P-values for each parameter (temperature, L/S ratio, temperature × temperature interaction, ratio × ratio interaction, temperature × ratio interaction). The P-values of them are important to comprehend which parameter is significant, or whether the interaction is significant or not. As the significant parameters are known, the insignificant parameter(s) can be removed from the regression equation, and therefore simplifying the regression equation.
Response 12: The P-values for each parameter are added as shown in Table 2
Table 2. Parameters of regression model for actual factors.
Parameters |
Total phenolics |
Total flavonoids |
Procyanidin |
DPPH |
FRAP |
|||||||
Coefficient |
p-value |
Coefficient |
p-value |
Coefficient |
p-value |
Coefficient |
p-value |
Coefficient |
p-value |
|||
Model |
|
|
|
|
|
|
||||||
Constant |
634.08 |
|
266.69 |
|
82.81 |
|
49.64 |
|
2497.37 |
|
||
Temperature |
-22.59 |
<0.001 |
-10.82 |
<0.001 |
-2.08 |
<0.001 |
-2.64 |
<0.001 |
-92.62 |
<0.001 |
||
Ratio |
-1.33 |
<0.001 |
3.01 |
0.001 |
-1.04 |
0.617 |
8.91 |
<0.001 |
15.31 |
<0.001 |
||
Temperature2 |
0.22 |
<0.001 |
0.11 |
<0.001 |
0.02 |
0.046 |
0.03 |
<0.001 |
0.88 |
<0.001 |
||
Ratio2 |
0.14 |
0.006 |
|
|
-0.05 |
0.186 |
|
|
|
|
||
Temp ´ Ratio |
-0.06 |
0.002 |
-0.05 |
<0.001 |
-0.02 |
0.156 |
-0.02 |
<0.001 |
-0.27 |
<0.001 |
||
p-value |
< 0.0001 |
< 0.0001 |
< 0.0001 |
< 0.0001 |
< 0.0001 |
|||||||
R2 |
0.9785 |
0.9523 |
0.7005 |
0.9991 |
0.9871 |
|||||||
In most cases, insignificant terms were excluded to simplify the model. The exception was the procyanidin model, in which insignificant ratio-containing terms were included to retain higher R2.
Point 13: Table 2: Where are the regression equations? --> for (1) total phenolic, (2) flavonoids, (3) procyanidin, (4) DPPH, and (5) FRAP??
Response 13: The regression equations for (1) total phenolic, (2) flavonoids, (3) procyanidin, (4) DPPH, and (5) FRAP are added in the section “3.2 Response Surface Methodology (RSM) Analysis”
as follows:
Regression models for actual factors (where T was temperature and R was liquid-to-solid ratio) were obtained as followed:
Total phenolics = 634.08 – 22.59T – 1.33R + 0.22T2 + 0.14R2 – 0.06TR
Total flavonoids = 266.69 – 10.82T + 3.01R + 0.11T2 – 0.05TR
Procyanidin = 82.81 – 2.08T – 1.04R + 0.02T2 – 0.05R2 – 0.02TR
DPPH = 49.64 – 2.64T + 8.91T + 0.03T2 – 0.02TR
FRAP = 2497.37 – 92.62T + 15.31R + 0.88T2 – 0.27TR
Point 14: Table 2 and Table 3: For the regression equation, the parameters are in coded units (-1, 0, +1) or in real values (such as temperature 40, 50, 60 °C, and not -1, 0, +1, respectively)?
Response 14:
Table 2: The parameters are in real values as indicated in the table title.
Table 3: The data presented in the table was only response variables, which had no coded units.
Point 15: Line 293-346: Please split this very long paragraph (takes up the whole page) to 4-6 shorter paragraphs.
Response 15: This long paragraph is splited into 5 paragraphs as follows:
In this research,….
In the case of freeze drying,….
Regarding the freeze-drying method,….
The antioxidant activity of the sample….
Although freeze drying is considered….
Point 16: Line 368-391: Please split this long paragraph to two shorter ones.
Response 16: This long paragraph is splited into 2 shorter paragraphs as follows:
The storability and eventual quality of GSS extract….
Procyanidins may be lost during storage….
Point 17: Line 378: Change “Contrariwise” to “In contrast”, or “On the contrary”, or “On the other hand”.
Response 17: The word of “Contrariwise” is changed to “In contrast”
Point 18: Line 392-423: Please split this long paragraph to two shorter ones.
Response 18: This long paragraph is splited in to 2 shorter paragraphs as follows:
In comparing the data to….
Storage temperature has been found….
Point 19: Line 454: Scientific names must be written in italic.
Response 19: Scientific names is corrected to “Phaseolus vulgaris”
Point 20: Line 513: Scientific names must be written in italic.
Response 20: Scientific names is corrected to “Acca sellowiana”
Point 21: Line 545: Scientific names must be written in italic. Write the genus Anemopsis with uppercase A
Response 21: Scientific names of “anemopsis californica” is corrected to “Anemopsis californica”
Kind regards,

Reviewer 2 Report
The article entitled “Optimization of Ultrasonic-Assisted Bioactive Compound Extraction from Green Soybean (Glycine max L.) and the Effect of Drying Methods and Storage Conditions on Procyanidin Extract” describes the optimization of the UAE extraction conditions (water to sample ratio and temperature) of bioactive compounds from green soybean seeds using RSM. In addition, the effects of various drying techniques (freeze-drying, hot air drying, spray drying, and vacuum drying) and storage conditions were compared based on procyanidins of green soybean extract.
In general, the article clearly presents the research done. However, some questions emerge that must be clarified/corrected:
- - Numeration of the analysis such as total phenolic contents, flavonoid contents, etc. (in the section) 2.6.1 is not appropriate
- The same comment is for section 2.6.2
- You applied four different methods to dry GSS extract to assess the impact of different drying methods on procyanidin content, antioxidant properties, and energy consumption. Why you left out monitoring the total polyphenol and flavonoid content?
- You can really upgrade your investigation using kinetic degradation models for the storage stability study.
I can suggest major revision before accepting the manuscript.
Author Response
Please see the attachment
Response to Reviewer 3 Comments
The authors thank Reviewer 3 for his/her kind and constructive comments to improve the quality and clarity of our manuscript.
Point 1: Numeration of the analysis such as total phenolic contents, flavonoid contents, etc. (in the section) 2.6.1 is not appropriate
Response 1: The subheading of 2.6.1.1, 2.6.1.2, and 2.6.1.3 in this section are added for the appropriate numeration.
Point 2: The same comment is for section 2.6.2
Response 2: The subheading of 2.6.2.1 and 2.6.2.2 in this section are added for the appropriate numeration.
Point 3: You applied four different methods to dry GSS extract to assess the impact of different drying methods on procyanidin content, antioxidant properties, and energy consumption. Why you left out monitoring the total polyphenol and flavonoid content?
Response 3: Authors need to focus on procyanidins, which are the potential bioactive compounds and present rather high concentration levels of green soybean extract as the latest publication in Foods 2022, 11(4), 588 (https://doi.org/10.3390/foods11040588). Process optimization for the production of procyanidin crude extract powder can be used to apply in pharmaceutical and food supplement industries. Procyanidins have attracted increasing attention due to their potential health benefits such as, anti-infectious, anti-inflammatory, and anti-aging.
Point 4: You can really upgrade your investigation using kinetic degradation models for the storage stability study.
Response 4: This study was conducted in order to evaluate the effect of storage conditions on procyanidins and antioxidant activities in GSS extract powder, which covers the ranges of storage temperature (25-45°C) and time (28 days) according to the objectives of the present study. Therefore, kinetic degradation models might be the only optional data for the storage stability study.
Kind regards,

Reviewer 3 Report
The submitted article “Optimization of Ultrasonic-Assisted Bioactive Compound Extraction from Green Soybean (Glycine max L.) and the Effect of Drying Methods and Storage Conditions on Procyanidin Extract” is well written, with proper methodology and results are very well justified with reasons and similar studies. I recommend major revision.
Abstract
Line 27: “soybeans” replace with “soybean”.
Line 32: Use correct symbol of degree “°”.
Line 38: “pow-der”, correct spelling.
Introduction
Line 51: Include reference.
Line 58: Enlist the benefits.
Line 70: “complicated working procedure”, explain why heating reflux extraction and homogenate extraction methods are complicated. Give reference for this.
Line 72: “accelerates oxidation of the extract”, Give reference.
Line 73: Write reference.
Write some of the earlier similar studies from recent past explaining how USA extraction technique is advantageous over traditional method.
Materials and Methods
Line 102: mention temperature.
Line 108: “different liquid-to-solid ratio”, which enzyme/solvent was used?
Line 111: Mention centrifugation RPM. Also write correct degree symbol. Follow this throughout the manuscript.
Results and Discussion
Authors have not included mathematical model obtained from Response Surface Methodology.
Table 2: Is there only one constant value in the model? Mention which type of equation was best fit.
Line 302 – 304: This is contrary to general effect of drying temperature and time on nutrients. Explain what is the special case in phytochemical concentrations and antioxidant properties of GSS?
Line 308: “freeze drying and hot air had the lowest”, sentence is incomplete.
Drying: Depending upon quality of product and drying energy requirement, which drying method is suggested from the study?
Storage study: Since most of the qualities are well retained at 25°C storage temperature which was the lowest among the selected level of the temperatures, can further reduction in storage temperature result in further maintenance of the quality parameters. Best storage temperature is not clear from this study. Why not to conduct one experiment at refrigeration condition?
Author Response
Response to Reviewer 2 Comments
The authors thank Reviewer 2 for his/her kind and constructive comments to improve the quality and clarity of our manuscript.
Abstract
Point 1: Line 27: “soybeans” replace with “soybean”.
Response 1: We made a correction at Line 27 from “soybeans” to “soybean”
Point 2: Line 32: Use correct symbol of degree “°”.
Response 2: We rechecked and changed the proper degree sign °C throughout the manuscript.
Point 3: Line 38: “pow-der”, correct spelling.
Response 3: We made a correction at Line 38 from “pow-der” to “powder”
Introduction
Point 4: Line 51: Include reference.
Response 4: The reference of Leksawasdi et al. (2022) https://doi.org/10.3390/foods11040588 is added in the sentence.
Point 5: Line 58: Enlist the benefits.
Response 5: The additional information is given as follows:
….as a result of their numerous benefits, such as anti-aging and anti-inflammatory.
Point 6: Line 70: “complicated working procedure”, explain why heating reflux extraction and homogenate extraction methods are complicated. Give reference for this.
Response 6: Further explanation and references of “complicated working procedure” by using reflux extraction and homogenate extraction methods is given as follows:
“However, it has been observed that the primary drawbacks of heating reflux extraction and homogenate extraction are the long extraction durations and low yields owing to re-peated distillation, which prolongs the heating time and promotes oxidation of the extract [6,7]. Moreover, because of their toxicity, volatility, and flammability, organic solvents are troublesome for procyanidin extraction [5].”
Point 7: Line 72: “accelerates oxidation of the extract”, Give reference.
Response 7: The additional reference is given as follows:
… and promotes oxidation of the extract [6,7]…
Point 8: Line 73: Write reference. Write some of the earlier similar studies from recent past explaining how USA extraction technique is advantageous over traditional method.
Response 8:
- The additional reference is given as follows:
“…Moreover, because of their toxicity, volatility, and flammability, organic solvents are troublesome for procyanidin extraction [5].”
- The additional information to explain the advantage of UAE technique over the traditional method is given as follows:
“Saifullah et al. [9] compared the extraction efficiency of UAE technique with conventional shaking water. The UAE technique was found to be more efficient in the extraction of TPC, TFC, proanthocyanidins, and antioxidant properties from lemon scented tea tree (Leptospermum petersonii) leaves. The TPC value using UAE was significantly higher (15.27%) than the value obtained for shaking water bath extraction.”
Materials and Methods
Point 9: Line 102: mention temperature.
Response 9: The temperature range of a refrigerator is added as follows:
“…and stored at a refrigerated temperature (3-5°C) until further analysis.”
Point 10: Line 108: “different liquid-to-solid ratio”, which enzyme/solvent was used?
Response 10: The sentence is added information as follows:
“Extractions using water as solvent were carried out in an ultrasonic probe…”
Point 11: Line 111: Mention centrifugation RPM. Also write correct degree symbol. Follow this throughout the manuscript.
Response 11: The unit for centrifugation is changed from g-force (RCF) to RPM as follows:
“After extraction, the mixtures were centrifuged at 5,000 RPM for 15 min.”
In addition, we rechecked and changed the proper degree sign °C throughout the manuscript.
Results and Discussion
Point 12: Authors have not included mathematical model obtained from Response Surface Methodology.
Response 12: Regression models for actual factors (where T was temperature and R was liquid-to-solid ratio) were obtained as follows:
Total phenolics = 634.08 – 22.59T – 1.33R + 0.22T2 + 0.14R2 – 0.06TR
Total flavonoids = 266.69 – 10.82T + 3.01R + 0.11T2 – 0.05TR
Procyanidin = 82.81 – 2.08T – 1.04R + 0.02T2 – 0.05R2 – 0.02TR
DPPH = 49.64 – 2.64T + 8.91T + 0.03T2 – 0.02TR
FRAP = 2497.37 – 92.62T + 15.31R + 0.88T2 – 0.27TR
Point 13: Table 2: Is there only one constant value in the model? Mention which type of equation was best fit.
Response 13: Constant value was the first term in the regression model which had no variables, as shown in the added equation.
Point 14: Line 302 – 304: This is contrary to general effect of drying temperature and time on nutrients. Explain what is the special case in phytochemical concentrations and antioxidant properties of GSS?
Response 14: GSS is abundant in flavonoids, which are mainly present in glycoside forms such as procyanidins, quercetin, glycitein, and daidzein. Moreover, glycosylated flavonoids are more resistant to heat treatment than aglycon flavonoids. Hence, the use of high temperatures in the spray drying method does not cause a drastic degradation of procyanidins, contributing to the high antioxidant activity of the obtained powders. In terms of time consumption for the drying of samples, freeze drying is considered a slow drying rate and a long drying time, which might contribute to unfavourable conditions for phenol composition, leading to destruction of the compounds. Some loss of bioactive compounds and volatile substances occurred mainly during the sublimation stage of ice. In the sublimation phase, compounds with a vapour pressure higher than water molecules are excluded and evaporated from the frozen materials when the sample matrix exceeds its glass transition temperature [26], leading to lower antioxidant activity when compared to the spray drying method.
The explanation in detail is given in section "3.3 Effect of Drying Methods on Procyanidin Content and Antioxidant Properties of GSS Extract " with the following sentences:
“However, GSS is abundant in flavonoids which are mainly present in glycoside forms such as procyanidins, quercetin, glycitein, and daidzein. Moreover, glycosylated flavonoids are more resistant to heat treatment than aglycon flavonoids. Hence, the use of high temperatures in the spray drying method does not cause a drastic degradation of procyanidins, contributing to the high antioxidant activity of the obtained powders [1,28].”
Point 15: Line 308: “freeze drying and hot air had the lowest”, sentence is incomplete.
Response 15: The sentence was rephrased to “…lemon pomace dried under vacuum had the higher total flavonoid content than freeze-drying and hot air-drying techniques.”
Point 16: Drying: Depending upon quality of product and drying energy requirement, which drying method is suggested from the study?
Response 16: The authors stated in the conclusion that “spray drying was the most effective in terms of the highest concentration level of procyanidins and antioxidant activity, as well as the lowest energy consumption. Hence, spray drying should be considered as a cost-effective drying method for large-scale production with the added benefit of maximum GSS extract recovery.”
Point 17: Storage study: Since most of the qualities are well retained at 25°C storage temperature which was the lowest among the selected level of the temperatures, can further reduction in storage temperature result in further maintenance of the quality parameters. Best storage temperature is not clear from this study. Why not to conduct one experiment at refrigeration condition?
Response 17: We need to assure that the final product (GSS powders) could be stable at the ambient temperature (25-30°C) including higher temperatures. In addition, these temperature levels might be proper for industrial application and cost efficiency in tropical zone countries.
Kind regards,

Round 2
Reviewer 1 Report
Review of foods-1769655-v2
The authors have addressed most of the issues previously raised. However, please revise the scientific in the Reference 38: Anemopsis californica --> in italic, started with uppercase A.
Author Response
Please see the attachment.
Response to Reviewer 1 Comments
The authors thank Reviewer 1 for his/her kind and constructive comments to improve the quality and clarity of our manuscript.
Point 1: Please revise the scientific in the Reference 38: Anemopsis californica --> in italic, started with uppercase A.
Response 1: We made a correction of the scientific in the Reference 38 from “anemopsis californica” to “Anemopsis californica”.
Kind regards,

Reviewer 2 Report
I suggest Accept in present form.
Author Response
Response to Reviewer 2 Comments
The authors thank Reviewer 2 for his/her constructive comments to improve the quality and clarity of the manuscript and accept the present form of our manuscript.
Kind regards,